# PowerGPT: Foundation Model for Power Systems

## Abstract

We propose a foundation model, namely PowerGPT , to model electricity time series (ETS) data, which learns generic representations of load and electricity consumption data by pre-training, providing a large-scale, off-the-shelf model for power systems. PowerGPT is the largest model in the field of power systems and is pre-trained on a large-scale ETS data including load and electricity consumption data. The design of PowerGPT is to capture long-term temporal dependency and hierarchical correlation from massive ETS data, providing information that spans from the fine-grained to coarse-grained scales. As a foundation model, PowerGPT achieves SOTA performance on various downstream tasks in power systems (i.e. forecasting, missing value imputation, and anomaly detection), showing the generalization ability to a wide range of tasks. The low-resource label analysis further illustrates the effectiveness of our pre-training strategy. In addition, we explore the effect of model size to show that a larger-scale model with a higher capacity can lead to performance improvements.

## 1 Introduction

Electricity time series (ETS) data, whose volume has recently surged due to the advanced power systems called smart grid (Fang et al., 2011). This abundance of data has paved the way for diverse applications in power systems, including demand-side management (Palensky & Dietrich, 2011), grid stability (Arzamasov et al., 2018) and consumer behavior analysis (Zhou & Yang, 2016), etc. Meanwhile, these applications have spawned various tasks as shown in Fig. 1(b), such as electricity load/consumption forecasting (Singh et al., 2012; Chandramitasari et al., 2018), missing value imputation (Lian et al., 2021), as well as electricity theft (Hu et al., 2020) and elderly living alone detection (Zhang et al., 2022a). Given the large amount of data and diverse downstream tasks, the exploration of how to effectively model ETS data for these tasks can bring improved economic efficiency and adhere to low-carbon principles.

Currently, studies for modeling ETS data can be grouped into three research lines, including traditional statistics based methods (de Assis Cabral et al., 2017; Attar et al., 2022), machine learning based methods (Kim et al., 2017; Wang et al., 2021) and deep learning based methods (Hasan et al., 2019; Hu et al., 2020; Lian et al., 2021). But most of them rely heavily on labeled data at scale, making it infeasible and expensive to obtain in power systems. Moreover, in the face of a variety of downstream tasks, it's inefficient to re-train a tailored model for each specific downstream task. Recently, many research about general time series pre-training approaches has appeared in other domains (e.g., weather, traffic flow, exchange rates, etc), such as PatchTST (Nie et al., 2023), TS2Vec (Yue et al., 2022), CoST (Woo et al., 2022), etc. They employ the "pre-training then

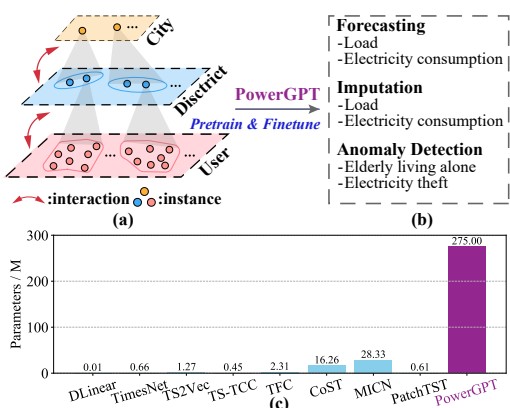

Figure 1: (a) The hierarchical structure of ETS data. (b) The diverse downstream tasks in power systems, and PowerGPT pre-train on ETS data, then finetune for various tasks. (c) The model scale of existing time series self-supervised models.

finetuning" paradigm, where the model is initially pre-trained on unlabeled time series to obtain a generic representation, and then subsequently finetuned for a specific downstream task. However, existing power systems related works still maintain a large research gap in modeling ETS data with this paradigm.

Nowadays, the "pre-training then finetuning" paradigm is also empolyed by large foundation models in CV/NLP (Devlin et al., 2018; Gao et al., 2020; He et al., 2022). These models provide robust generic representations through pre-training with massive unlabeled data, and demonstrate remarkable performance on downstream tasks through fine-tuning with a small amount of labeled data (He et al., 2019). Given the massive unlabeled data and diverse downstream tasks in power systems, there is an urgent imperative to develop a foundation model with generalization capabilities to unify these downstream tasks by leveraging these massive data.

In our scenario, the ETS data typically follows a naturally complex geographical hierarchy according to Yang et al. (2015); Pang et al. (2018). A vivid illustration can be observed in Fig. 1(a), a city's ETS can be disaggregated into districts' data through the administrative divisions, which are further disaggregated into users' data in this districts. When encountering the complex hierarchy in massive ETS data, modeling ETS data entails the careful consideration of several critical factors:

**(1) ETS data exhibits long-term temporal dependency.** Since ETS data displays distinct patterns across days, weeks, seasons, and years, long-term temporal dependency is crucial in modeling ETS data. **(2) Heterogeneity patterns of different instances in ETS data.** The heterogeneity of electricity consumption patterns refers to the significant variations among different instances of electricity consumers, such as individual users, districts, cities, and other entities. By considering this heterogeneity, it is possible to better represent real-world scenarios and accurately capture the variations in ETS data. **(3) Hierarchical correlation across different instances in ETS data.** Given the natural hierarchical structure of electricity consumption data, interactions occur between fine-grained and coarse-grained. Fine-grained information of user offers insights into individual factors and micro trends, while coarse-grained information of city reveals broader factors and macro trends. Consequently, by considering the hierarchical relationship to model ETS data, the representations can capture more comprehensive information and be closer to the real scene. To the best of our knowledge, no existing work on ETS data considers all the three critical factors simultaneously.

To address these considerations above, we propose a foundation model for ETS data named **Power Generic Pre-trained Transformer (PowerGPT )**. The design of our model takes three key factors (long-term temporal dependency, heterogeneity patterns of different instances and hierarchical correlation) for electricity data modeling into account. Moreover, PowerGPT contains more than 270M parameters and is pre-trained on a large-scale ETS data with 1TB , which can be adapted to accomplish various downstream tasks. Compared to other existing methods for ETS data, PowerGPT can achieve better performance with far fewer labeled samples, showing the great benefit of our work in power systems scenarios.

To sum up, the main contributions of our work comprise:

1. We propose a foundation model for power systems named PowerGPT , which is the largest model in power systems (shown in Fig. 1(c)) and pre-trained on a large-scale ETS data provided by State Grid [1], providing a generic model for power systems.

2. To the best of our knowledge, PowerGPT is the first to date that attends long-term temporal dependency, heterogeneity patterns of different instances and captures hierarchical correlation across different instances.

3. Extensive experiments show that PowerGPT generalizes well to various downstream tasks, showing a great potential in ETS data modeling. Further analysis illustrates the effectiveness of the large-scale pre-trained model, demonstrating the application value of our work.

---

[1] http://www.sgcc.com.cn/ywlm/index.shtml

## 2 RELATED WORK

### 2.1 SELF-SUPERVISED PRE-TRAINING

Large-scale model based on self-supervised pre-training has become more and more significant in both industrial and academic domains due to the versatility and impressive performance. It initially developed and matured in the fields of computer vision (He et al., 2022) and natural language processing (Devlin et al., 2018; Gao et al., 2020). And recently it has made tremendous progress in time series such as WavLM (Chen et al., 2022) for audio, Pangu (Bi et al., 2023) for weather forecasting. Self-supervised pre-training in time series is typically classified into two paradigms: contrastive learning and mask modeling. The objective of contrastive learning is to learn representation by pushing positive pairs closer and negative pairs far from each other in the embedding space (Jaiswal et al., 2020). TS2Vec (Yue et al., 2022) proposes contextual consistency for positive pair selection. Afterward, CoST (Woo et al., 2022) extracts the trend and seasonal feature representations, and takes advantage of both time and frequency domain contrastive loss to encourage discriminative seasonal representation. TS-TCC (Eldele et al., 2021) presents a temporal contrastive module based on cross-view prediction. And TF-C (Zhang et al., 2022b) applies time-frequency consistency for embedding time-based and frequency-based neighbors. In mask modeling, The core idea is to recover the masked content from the unmasked part. TST (Zerveas et al., 2021) trys to predict the masked points in time series to learn representation with the remaining points, using the denosing autoencoder. To extract the contextual semantic information, PatchTST (Nie et al., 2023) and TimeMAE (Cheng et al., 2023) achieve masking at the series-level. However, despite the existence of numerous methods for self-supervised pre-training of time series, the research on large-scale models specifically designed for power systems in time series remains relatively sparse.

### 2.2 POWER SYSTEM RELATED TASKS

In the real world, the typical power system tasks related to time series are divided into three categories: forecasting, imputation, and anomaly detection.

**Forecasting** : It refers to the prediction of future series demand over a given time period and includes electricity consumption forecasting (Gonzalez-Briones et al., 2019), load forecasting (Alfares & Nazeeruddin, 2002; Negnevitsky et al., 2009). It is a vital task in power system planning, operation, and decision-making processes. Torres et al. (2022) design a deep LSTM network for the Spanish electricity consumption forecasting. And SVR (Hong, 2009), CNN-BiGRU (Niu et al., 2022) are applied to address load forecasting problem.

**Missing Value Imputation** : Imputation in power systems is to estimating or predicting the missing values based on the available neighbor information within the power system datasets, ensuring that the datasets used for analysis, modeling, and decision-making are complete and reliable. (Kim et al., 2017) uses a learning-based adaptive imputation method based on KNN algorithm for filling missing power data in an energy system. (Moghaddass & Wang, 2017) impute the missing value in the smart grid system by modeling the data using probabilistic distributions. For load data, (Jeong et al., 2021) present a mixture factor analysis(MFA) method for estimating missing values in building electric load data. And (Kamisan et al., 2020) develop an imputation model for incomplete load data based on seasonality and orientation of the missing value.

**Anomaly Detection** : This task is defined as the process of identifying abnormal or unusual behavior in the operation, performance, or data presentation of the power system. It involves monitoring and analyzing various signals, measurements, and data streams within the power system to detect deviations from normal operating conditions or expected patterns (Yan & Wen, 2021), which involves electricity theft detection and elder living alone detection. Jokar et al. (2015) design consumption pattern based energy theft detector, which leverages the predictability derived from customers' normal and malicious consumption patterns. Hasan et al. (2019); Adil et al. (2020) build a CNN-based LSTM model for electricity theft detection in smart grid Systems. And Karim et al. (2019) propose LSTM-FCNs to classify the time series and detect the anomaly. Zhang et al. (2022a) construct an unsupervised guardianship model for the elderly living alone through the characteristics of power consumption curve. But so far, there does not yet exist a unified model that can effectively solve all the tasks mentioned above.

## 3 METHODOLOGY

**Model Overview.** As previously mentioned, we propose PowerGPT to capture long-term temporal dependency and hierarchical correlation in ETS data. PowerGPT consists of two main modules: temporal encoder and hierarchical encoder (shown in Fig. 2). The ETS data is initially constructed to hierarchical graph. The temporal encoder is responsible for encoding $N_p$ consecutive patches of ETS into temporal representations, which specifically focus on modeling long-term temporal dependencies. Furthermore, the hierarchical encoder encodes temporal representations with the same timestamps from all hierarchical levels, capturing the underlying hierarchical correlations by leveraging information spanning from fine-grained (individual users) to coarse-grained (districts, cities and province).

During pre-training, the patched ETS is randomly masked using a masking strategy that involves temporal masking and hierarchical masking. Next, the masked patches are linearly mapped to the latent representations, where the positional information is added, and the masked portion is replaced with learnable tokens. After concatenated with the instance indicator, the representations are encoded into temporal representations through multiple Transformer encoders and then inputted into a hierarchical encoder, which maps the representations to reconstruct the original time series. The details of PowerGPT are described below.

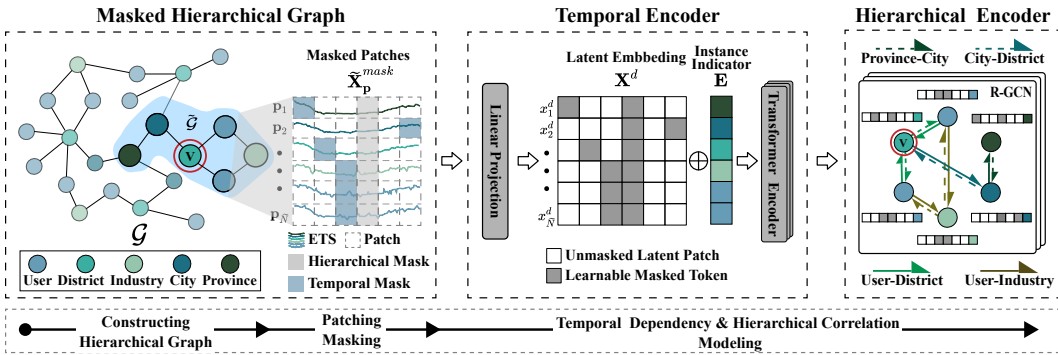

Figure 2: The pre-training framework of PowerGPT. ETS data is first constructed to a hierarchical graph $\mathcal{G}$ based on the subordinate relations among instances. Then a subgraph $\widetilde{\mathcal{G}}$ of a target node **v** will be sampled, whose ETS will be divided into patches. Next, we randomly mask a subset of patches by empolying temporal mask and hierarchical mask to get the input $\widetilde{\mathbf{X}}_{\mathbf{p}}^{mask}$. After a linear projection layer, the input will be mapped to the latent embbeding $\mathbf{X}^d$, where the learnable token will be used to replace the masked portion of patches and concatenate instance learnable indicator **E**. Subsequently, the augmented latent embedding is fed into multiple Transformer encoders to model long-term temporal dependency. Finally, the hierarchical encoder models hierarchical correlation to obtain the final generic representations by R-GCN (Schlichtkrull et al., 2018).

**Hierarchical Graph Construction.** ETS data contains a quantity of time series and the condition for the existence of a relationship between two time series is defined based on the naturally existing hierarchical structure. This relationship is established when their corresponding instances have a subordinate relationship, such as user belonging to district and district belonging to city. Mathematically, we transform it into a hierarchical graph denoted as $\mathcal{G} = (\mathcal{V}, \mathcal{U}, \mathcal{E}, \mathcal{R})$, where $\mathcal{V}$ is the vertex set, $\mathcal{U}$ is the attribute associated with each vertex (e.g, instance type), $\mathcal{E}$ is the corresponding directed edge set, and $\mathcal{R}$ is the relationship set among edges. For a vertex $\boldsymbol{v}_i \in \mathcal{V}$, we denote its neighbors in relation $r$ as $\mathcal{N}_i^r$, where $r \in \mathcal{R}$ is the relation type.

**Patching.** Considering the inherent long-term temporal dependency and diverse patterns in ETS data, we divide ETS into patches for the following reasons: **(1)** to enhance locality and extract semantic information; **(2)** to reduce computational complexity and memory usage; and **(3)** to obtain a longer temporal dependency (Nie et al., 2023). Specifically, let $\mathbf{X} = [\boldsymbol{x}_1, \boldsymbol{x}_2, \cdots, \boldsymbol{x}_N]^{\mathrm{T}} \in \mathbb{R}^{N \times T_w}$ as the ETS of all instances in a sliding window, where $T_w$ denotes the length of sliding window. And we divide $\boldsymbol{x}_i$ with a length of $P$ and a stride of $S$, which generates a series of patches, denoted as $\mathbf{p}_i \in \mathbb{R}^{N_p \times P}$, where $N_p = \lceil \frac{T_w - P}{S} \rceil + 1$ represents the number of patches in sliding window. $\forall \boldsymbol{x}_i \in \mathbf{X}$, we apply this patching process on it, then we obtain $\mathbf{X_p} = [\mathbf{p}_1, \mathbf{p}_2, \cdots, \mathbf{p}_N]^{\mathrm{T}} \in \mathbb{R}^{N \times N_p \times P}$ as the patched $\mathbf{X}$.

**Randomly Masking.** The concept of using masked autoencoders for pre-training has been extensively applied in CV (Bao et al., 2021; He et al., 2022) and NLP (Devlin et al., 2018). Another significant concept for pre-training is contrastive learning, but some research (Tian et al., 2020; Zheng et al., 2023) indicates that the construction of positive and negative pairs heavily depends on specific time series characteristics, which limits the generalization across diverse types of time series data. Motivated by these, we utilize masked autoencoders as the backbone of our model.

Moreover, the traditional time series masking strategy focuses on modeling temporal dependency, but neglects the influence of hierarchical correlation. In our scenario, different levels of information can interact with others. Specifically, at given timestamps, the unmasked levels provide auxiliary information to the masked levels, thereby simplifying the modeling process of ETS, weakening the expressive ability of the model.

To effectively model long-term temporal dependency and hierarchical correlation in ETS data respectively, we propose temporal masking and hierarchical masking, and their corresponding mask matrix denoted as $\widetilde{\mathbf{M}}_t$ and $\widetilde{\mathbf{M}}_h$. As shown in the left part of Fig. 2, temporal masking refers to the masked and unmasked patches of different nodes can overlap at temporal axis. This means $\oplus_{i=1}^{\widetilde{N}} \widetilde{\mathbf{M}}_t[i] \neq \vec{\mathbf{0}}$, where $\oplus$ is logical XOR operation. In contrast, hierarchical masking indicates that the masked positions and unmasked patches of different nodes cannot overlap at temporal axis, which means $\oplus_{i=1}^{\widetilde{N}} \widetilde{\mathbf{M}}_h[i] = \vec{\mathbf{0}}$. Then, we element-wise compute $\widetilde{\mathbf{M}}_t \vee \widetilde{\mathbf{M}}_t$ to form mask matrix $\widetilde{\mathbf{M}}$, where $\vee$ is logical OR operation.

Finally, we sample a subgraph $\widetilde{\mathcal{G}}$ of a target node with $\widetilde{N}$ nodes, and its corresponding patched ETS data is $\widetilde{\mathbf{X}}_{\mathbf{p}} \in \mathbb{R}^{\widetilde{N} \times N_p \times P}$. Then we use the mask matrix $\widetilde{\mathbf{M}} \in \mathbb{R}^{\widetilde{N} \times N_p}$ to mask $\widetilde{\mathbf{X}}_{\mathbf{p}}$ to obtain $\widetilde{\mathbf{X}}_{\mathbf{p}}^{mask}$.

**Temporal Encoder.** In our approach, we employ multiple Transformer encoders to map ETS data to corresponding temporal representations. Firstly, $\widetilde{\mathbf{X}}_{\mathbf{p}}^{mask}$ is mapped to the latent space of dimension $D$ via a linear projection $\mathbf{W}_{in} \in \mathbb{R}^{P \times D}$, then learnable mask token is used to replace with the masked portion to form latent embedding $\mathbf{X}^d \in \mathbb{R}^{\widetilde{N} \times N_p \times D}$. Given the complex electricity consumption behavior among various instances, we propose a learnable instance indicator set $\{\mathbf{e}_i\}_{i=1}^{N_{instance}}$ among all instances, where $\mathbf{e}_i \in \mathbb{R}^D$. Next, we retrieval the corresponding instance indicator for each nodes in $\widetilde{\mathcal{V}}$ to form instance indicator matrix $\mathbf{E} \in \mathbb{R}^{\widetilde{N} \times D}$, and then concatenate with latent embedding $\mathbf{X}^d$ to form $\mathbf{Z}^d$. Finally, each $\mathbf{z}_i$ of nodes will be fed into multiple Transformer encoders. Afterwards it generates the temporal representations denoted as $\mathbf{H}_i \in \mathbb{R}^{\widetilde{N} \times (N_p+1) \times D}$.

**Hierarchical Encoder.** To further enhance the temporal representation, we incorporate hierarchical correlation modeling, which takes into account both fine-grained user-related information and coarse-grained information from district, city, and province levels. By incorporating fine-grained user ETS information, more refined representations can be provided, particularly when dealing with a large number of users and significant variations in power consumption patterns. Meanwhile, coarse-grained ETS enables the capture of broader influencing factors and overall trends. For instance, city level may include city size, economic development levels, and industrial structures, while province level may encompass climatic conditions, demographic data, and policy impacts (Xu & Jiao, 2021). These coarse-grained details offer a more macro perspective, aiding model in comprehending overall ETS trends and fluctuations.

Recently, GNNs (Wu et al., 2020) have been utilized to fuse each time series information with its neighboring time series, resulting in an improved understanding of related patterns within each series. Motivated by these research, we employ R-GCN to integrate information from different hierarchies of instances. Specifically, we define the following propagation model for calculating the forward-pass update of a node denoted by $v_i$ in our hierarchical graph:

$$\mathbf{H}_i^{(l+1)} = \sigma \left( \sum_{r \in \mathcal{R}} \sum_{j \in \mathcal{N}_i^r} \frac{1}{c_{i,r}} \mathbf{W}_r^{(l)} \mathbf{H}_j^{(l)} + \mathbf{W}_0^{(l)} \mathbf{H}_i^{(l)} \right) \quad (1)$$

In this equation, $\mathbf{H}_i^{(l+1)}$ represents the updated representation of node $i$ at layer $l+1$, $\sigma(\cdot)$ denotes the activation function, $\mathcal{R}$ represents the set of all relation types in the graph, $\mathcal{N}_i^r$ denotes the set of

Table 1: Full results for the load forecasting task. We compare extensive competitive models under different forecasting horicontal (24, 96, 336, 720). The input sequence length is set to 256. *Avg* is averaged from all four prediction lengths.

| Model | | PowerGPT | | PatchTST | | TS2Vec | | CoST | | TS-TCC | | TFC | | DLinear | | MICN | | LSTM | |
|---|---|---|---|---|---|---|---|---|---|---|---|---|---|---|---|---|---|---|---|
| Metric | | MSE | MAE | MSE | MAE | MSE | MAE | MSE | MAE | MSE | MAE | MSE | MAE | MSE | MAE | MSE | MAE | MSE | MAE |
| Exclusive | 24 | 0.310 | 0.347 | 0.622 | 0.551 | 0.297 | 0.345 | 0.593 | 0.574 | 0.550 | 0.532 | 0.572 | 0.504 | 0.596 | 0.458 | 0.570 | 0.443 | 0.831 | 0.606 |
| | 96 | 0.426 | 0.395 | 0.507 | 0.481 | 0.931 | 0.577 | 0.580 | 0.568 | 0.594 | 0.548 | 0.485 | 0.464 | 0.699 | 0.502 | 0.432 | 0.464 | 1.306 | 0.825 |
| | 336 | 0.429 | 0.383 | 0.541 | 0.501 | 0.478 | 0.443 | 0.659 | 0.590 | 0.666 | 0.604 | 0.516 | 0.470 | 0.768 | 0.536 | 0.753 | 0.526 | 1.617 | 0.884 |
| | 720 | 0.431 | 0.396 | 0.646 | 0.527 | 0.697 | 0.526 | 0.669 | 0.587 | 0.696 | 0.623 | 0.529 | 0.486 | 0.664 | 0.505 | 0.652 | 0.499 | 1.100 | 0.762 |
| | Avg | **0.399** | **0.380** | 0.579 | 0.515 | 0.600 | 0.473 | 0.625 | 0.580 | 0.627 | 0.577 | 0.525 | 0.481 | 0.682 | 0.500 | 0.602 | 0.483 | 1.214 | 0.769 |
| Public | 24 | 0.200 | 0.289 | 0.359 | 0.444 | 0.360 | 0.434 | 1.315 | 0.810 | 0.500 | 0.557 | 0.367 | 0.439 | 0.518 | 0.497 | 0.514 | 0.494 | 1.367 | 0.871 |
| | 96 | 0.316 | 0.357 | 0.377 | 0.450 | 0.843 | 0.633 | 0.469 | 0.522 | 0.460 | 0.531 | 0.390 | 0.457 | 0.679 | 0.569 | 0.162 | 0.266 | 1.315 | 0.829 |
| | 336 | 0.594 | 0.544 | 0.399 | 0.470 | 0.379 | 0.444 | 0.506 | 0.543 | 0.513 | 0.568 | 0.414 | 0.463 | 0.918 | 0.645 | 0.944 | 0.649 | 1.764 | 0.925 |
| | 720 | 0.263 | 0.341 | 0.500 | 0.498 | 0.726 | 0.572 | 0.547 | 0.556 | 0.524 | 0.577 | 0.412 | 0.471 | 0.684 | 0.553 | 0.698 | 0.554 | 1.111 | 0.766 |
| | Avg | **0.343** | **0.383** | 0.409 | 0.466 | 0.577 | 0.521 | 0.709 | 0.608 | 0.500 | 0.558 | 0.396 | 0.458 | 0.700 | 0.566 | 0.579 | 0.491 | 1.389 | 0.848 |
| Industry | 24 | 0.101 | 0.223 | 0.315 | 0.436 | 0.114 | 0.255 | 0.472 | 0.561 | 0.327 | 0.428 | 0.444 | 0.488 | 0.325 | 0.383 | 0.303 | 0.371 | 1.029 | 0.796 |
| | 96 | 0.167 | 0.269 | 0.269 | 0.394 | 0.723 | 0.584 | 0.354 | 0.483 | 0.354 | 0.457 | 0.234 | 0.367 | 0.473 | 0.476 | 0.233 | 0.351 | 1.311 | 0.907 |
| | 336 | 0.276 | 0.336 | 0.246 | 0.384 | 0.212 | 0.339 | 0.426 | 0.517 | 0.364 | 0.478 | 0.274 | 0.388 | 0.689 | 0.557 | 0.696 | 0.559 | 2.084 | 1.064 |
| | 720 | 0.210 | 0.309 | 0.447 | 0.469 | 0.564 | 0.503 | 0.481 | 0.531 | 0.392 | 0.502 | 0.301 | 0.411 | 0.541 | 0.473 | 0.554 | 0.479 | 1.164 | 0.831 |
| | Avg | **0.189** | **0.284** | 0.319 | 0.421 | 0.403 | 0.420 | 0.433 | 0.523 | 0.359 | 0.466 | 0.313 | 0.413 | 0.507 | 0.472 | 0.446 | 0.440 | 1.397 | 0.900 |
| District | 24 | 0.154 | 0.229 | 0.191 | 0.352 | 0.148 | 0.294 | 1.028 | 0.780 | 0.289 | 0.375 | 0.286 | 0.400 | 0.292 | 0.362 | 0.270 | 0.350 | 0.568 | 0.585 |
| | 96 | 0.144 | 0.238 | 0.202 | 0.340 | 0.848 | 0.648 | 0.342 | 0.461 | 0.313 | 0.428 | 0.211 | 0.345 | 0.471 | 0.472 | 0.124 | 0.264 | 1.624 | 1.005 |
| | 336 | 0.222 | 0.299 | 0.205 | 0.346 | 0.183 | 0.319 | 0.416 | 0.516 | 0.363 | 0.482 | 0.230 | 0.367 | 0.780 | 0.578 | 0.809 | 0.585 | 2.022 | 1.089 |
| | 720 | 0.189 | 0.270 | 0.286 | 0.373 | 0.567 | 0.488 | 0.450 | 0.527 | 0.363 | 0.483 | 0.316 | 0.421 | 0.524 | 0.448 | 0.535 | 0.447 | 1.232 | 0.878 |
| | Avg | **0.178** | **0.259** | 0.221 | 0.353 | 0.437 | 0.437 | 0.559 | 0.571 | 0.332 | 0.442 | 0.261 | 0.383 | 0.517 | 0.465 | 0.435 | 0.411 | 1.361 | 0.889 |
| City | 24 | 0.075 | 0.126 | 0.486 | 0.558 | 1.353 | 0.823 | 1.202 | 0.908 | 0.167 | 0.281 | 0.382 | 0.510 | 0.187 | 0.301 | 0.135 | 0.260 | 0.790 | 0.720 |
| | 96 | 0.055 | 0.209 | 0.257 | 0.435 | 1.092 | 0.824 | 0.710 | 0.696 | 0.377 | 0.545 | 0.148 | 0.311 | 0.385 | 0.448 | 0.163 | 0.296 | 1.027 | 0.868 |
| | 336 | 0.302 | 0.356 | 0.932 | 0.631 | 1.961 | 1.008 | 0.794 | 0.684 | 1.695 | 1.067 | 0.205 | 0.336 | 0.704 | 0.558 | 0.633 | 0.515 | 2.010 | 1.082 |
| | 720 | 0.724 | 0.574 | 0.703 | 0.609 | 0.642 | 0.600 | 0.693 | 0.635 | 1.021 | 0.796 | 1.307 | 0.835 | 0.476 | 0.433 | 0.464 | 0.423 | 0.936 | 0.773 |
| | Avg | **0.289** | **0.316** | 0.595 | 0.558 | 1.262 | 0.814 | 0.850 | 0.731 | 0.815 | 0.672 | 0.511 | 0.498 | 0.438 | 0.435 | 0.349 | 0.373 | 1.191 | 0.861 |
| Province | 24 | 0.174 | 0.219 | 0.559 | 0.623 | 2.709 | 1.054 | 1.089 | 0.785 | 0.175 | 0.284 | 0.514 | 0.534 | 0.105 | 0.233 | 0.091 | 0.228 | 0.348 | 0.573 |
| | 96 | 0.080 | 0.164 | 0.200 | 0.392 | 0.753 | 0.673 | 0.803 | 0.715 | 0.466 | 0.572 | 0.171 | 0.337 | 0.215 | 0.348 | 0.444 | 0.497 | 0.553 | 0.655 |
| | 336 | 0.688 | 0.408 | 1.510 | 0.698 | 2.811 | 1.052 | 1.104 | 0.776 | 2.240 | 1.120 | 0.180 | 0.314 | 0.338 | 0.412 | 0.309 | 0.386 | 1.063 | 0.850 |
| | 720 | 0.484 | 0.369 | 0.876 | 0.577 | 0.398 | 0.501 | 0.812 | 0.652 | 1.377 | 0.875 | 1.890 | 0.894 | 0.254 | 0.363 | 0.264 | 0.366 | 0.642 | 0.696 |
| | Avg | 0.357 | **0.290** | 0.786 | 0.572 | 1.668 | 0.820 | 0.952 | 0.732 | 1.064 | 0.713 | 0.689 | 0.520 | **0.228** | 0.339 | 0.277 | 0.369 | 0.651 | 0.694 |

neighboring nodes of node $i$ with relation type $r$. The normalization factor $c_{i,r}$ is typically defined as the square root of the degree of node $i$ with relation type $r$. The learnable weight matrix $\mathbf{W}_r^{(l)}$ is specific to relation type $r$ at layer $l$, while $\mathbf{H}_j^{(l)}$ represents the representation of neighboring node $j$ at layer $l$. Furthermore, $\mathbf{W}_0^{(l)}$ denotes a learnable self-loop weight matrix at layer $l$, and $\mathbf{H}_i^{(l)}$ represents the representation of the current node $i$ at layer $l$.

## 4 EXPERIMENTAL SETUP

The large amount of ETS data provided by State Grid will be divided into pre-triaining dataset and finetuning dataset without overlap for PowerGPT pre-training and finetuning (detailed in App. A). Moreover, we select latest SOTAs for time series modeling as our baselines (detailed in App. B). Furthermore, to verify the capacity of PowerGPT in modeling ETS data, we conduct extensive experiments on several downstream tasks (detailed in App. C).

## 5 EXPERIMENTAL RESULTS

**Overview** Our codes are available in https://anonymous.4open.science/r/PowerGPT-0152. PowerGPT achieves SOTA performance on various tasks with other baseline models in power systems (i.e. forecasting, missing value imputation, anomaly detection), showing the generalization ability to a broad range of tasks. We delve into more detailed comparisons of each task in the following paragraphs, where in all the tables we mark the best results are in **v**.

**Forecasting.** The result of load and electricity consumption forecasting in various forecasting horizontal are shown in Tab. 1 and Tab. 2, respectively. These results show that not only does PowerGPT achieve SOTA performance, but the results of PowerGPT -Freeze are also better than most baselines, showing the ability to capture long-term temporal dependency and hierarchical correlation of our model. Among the baselines, PatchTST performs well compared to other methods, mainly because it also adopts a patching strategy to model longer temporal dependency.

**Anomaly Detection.** We conduct anomaly detection experiment on two tasks: electricity theft detection and elderly living alone detection. Tab. **??** and Tab. **??** demonstrate that PowerGPT still achieves the best performance is anomaly detection, outperforming the advanced mask reconstruction based model PatchTST and constrastive learning based model CoST. The canonical Transformer

Table 2: Full results for the electricity forecasting task. We compare extensive competitive models under different forecasting horizontals. The input sequence length is set to 256. *Avg* is averaged from all four prediction lengths.

| Model | | PowerGPT | | PatchTST | | TS2Vec | | CoST | | TS-TCC | | TFC | | DLinear | | MICN | | LSTM | |
|---|---|---|---|---|---|---|---|---|---|---|---|---|---|---|---|---|---|---|---|
| Metric | | MSE | MAE | MSE | MAE | MSE | MAE | MSE | MAE | MSE | MAE | MSE | MAE | MSE | MAE | MSE | MAE | MSE | MAE |
| Exclusive | 12 | 0.350 | 0.398 | 0.372 | 0.427 | 0.460 | 0.489 | 0.551 | 0.560 | 0.658 | 0.646 | 0.849 | 0.642 | 0.449 | 0.489 | 0.630 | 0.602 | 0.632 | 0.675 |
| | 48 | 0.374 | 0.426 | 0.410 | 0.454 | 0.665 | 0.564 | 0.643 | 0.596 | 0.664 | 0.644 | 0.529 | 0.519 | 0.508 | 0.520 | 0.543 | 0.544 | 0.644 | 0.714 |
| | 96 | 0.441 | 0.466 | 0.584 | 0.568 | 0.480 | 0.500 | 0.745 | 0.656 | 0.664 | 0.641 | 0.705 | 0.604 | 0.436 | 0.467 | 0.432 | 0.464 | 0.743 | 0.779 |
| | Avg | **0.389** | **0.430** | 0.455 | 0.483 | 0.535 | 0.518 | 0.646 | 0.604 | 0.662 | 0.644 | 0.695 | 0.588 | 0.464 | 0.492 | 0.535 | 0.537 | 0.673 | 0.722 |
| Public | 12 | 0.154 | 0.245 | 0.177 | 0.272 | 0.184 | 0.312 | 0.239 | 0.373 | 0.404 | 0.511 | 0.315 | 0.399 | 0.254 | 0.385 | 0.229 | 0.356 | 0.708 | 0.775 |
| | 48 | 0.193 | 0.303 | 0.198 | 0.288 | 0.271 | 0.347 | 0.270 | 0.379 | 0.485 | 0.606 | 0.279 | 0.359 | 0.158 | 0.280 | 0.195 | 0.335 | 0.821 | 0.876 |
| | 96 | 0.166 | 0.279 | 0.275 | 0.397 | 0.196 | 0.316 | 0.422 | 0.511 | 0.427 | 0.543 | 0.323 | 0.417 | 0.152 | 0.251 | 0.162 | 0.266 | 0.993 | 1.046 |
| | Avg | **0.171** | **0.275** | 0.217 | 0.319 | 0.217 | 0.325 | 0.310 | 0.421 | 0.439 | 0.553 | 0.306 | 0.392 | 0.188 | 0.305 | 0.195 | 0.319 | 0.841 | 0.899 |
| Industry | 12 | 0.258 | 0.374 | 0.259 | 0.359 | 0.272 | 0.417 | 0.329 | 0.464 | 0.554 | 0.617 | 0.627 | 0.562 | 0.250 | 0.388 | 0.564 | 0.653 | 0.507 | 0.548 |
| | 48 | 0.253 | 0.371 | 0.229 | 0.352 | 0.451 | 0.460 | 0.399 | 0.487 | 0.512 | 0.613 | 0.369 | 0.448 | 0.246 | 0.385 | 0.314 | 0.444 | 0.553 | 0.601 |
| | 96 | 0.195 | 0.326 | 0.324 | 0.445 | 0.247 | 0.379 | 0.469 | 0.541 | 0.513 | 0.598 | 0.557 | 0.549 | 0.234 | 0.343 | 0.233 | 0.351 | 0.601 | 0.636 |
| | Avg | **0.235** | **0.357** | 0.271 | 0.385 | 0.323 | 0.419 | 0.399 | 0.497 | 0.526 | 0.609 | 0.518 | 0.520 | 0.243 | 0.372 | 0.370 | 0.483 | 0.554 | 0.595 |
| District | 12 | 0.107 | 0.250 | 0.166 | 0.300 | 0.200 | 0.379 | 0.240 | 0.427 | 0.317 | 0.472 | 0.764 | 0.698 | 0.209 | 0.370 | 0.471 | 0.611 | 0.494 | 0.555 |
| | 48 | 0.079 | 0.219 | 0.090 | 0.231 | 0.288 | 0.375 | 0.204 | 0.362 | 0.394 | 0.558 | 0.217 | 0.366 | 0.130 | 0.258 | 0.200 | 0.346 | 0.505 | 0.608 |
| | 96 | 0.070 | 0.206 | 0.179 | 0.334 | 0.141 | 0.286 | 0.290 | 0.411 | 0.323 | 0.489 | 0.441 | 0.478 | 0.121 | 0.265 | 0.124 | 0.264 | 0.541 | 0.659 |
| | Avg | **0.085** | **0.225** | 0.145 | 0.288 | 0.209 | 0.346 | 0.245 | 0.400 | 0.345 | 0.506 | 0.474 | 0.514 | 0.153 | 0.298 | 0.265 | 0.407 | 0.514 | 0.607 |
| City | 12 | 0.061 | 0.187 | 0.186 | 0.306 | 0.220 | 0.403 | 0.234 | 0.413 | 0.247 | 0.416 | 0.844 | 0.688 | 0.177 | 0.335 | 0.653 | 0.720 | 0.192 | 0.202 |
| | 48 | 0.076 | 0.213 | 0.062 | 0.194 | 0.427 | 0.466 | 0.180 | 0.333 | 0.283 | 0.453 | 0.259 | 0.372 | 0.148 | 0.265 | 0.213 | 0.349 | 0.346 | 0.376 |
| | 96 | 0.058 | 0.187 | 0.162 | 0.298 | 0.111 | 0.251 | 0.230 | 0.367 | 0.230 | 0.397 | 0.538 | 0.512 | 0.147 | 0.281 | 0.163 | 0.296 | 0.446 | 0.490 |
| | Avg | **0.065** | **0.196** | 0.137 | 0.266 | 0.253 | 0.374 | 0.214 | 0.371 | 0.253 | 0.422 | 0.547 | 0.524 | 0.158 | 0.293 | 0.343 | 0.455 | 0.328 | 0.356 |
| Province | 12 | 0.135 | 0.278 | 0.369 | 0.465 | 0.231 | 0.379 | 0.327 | 0.467 | 0.237 | 0.403 | 2.135 | 1.090 | 0.194 | 0.349 | 0.779 | 0.744 | 0.145 | 0.192 |
| | 48 | 0.150 | 0.288 | 0.206 | 0.388 | 1.271 | 0.802 | 0.414 | 0.538 | 0.178 | 0.295 | 0.582 | 0.548 | 0.230 | 0.304 | 0.248 | 0.284 | 0.202 | 0.245 |
| | 96 | 0.127 | 0.258 | 0.462 | 0.566 | 0.332 | 0.493 | 0.368 | 0.494 | 0.205 | 0.327 | 1.360 | 0.881 | 0.443 | 0.510 | 0.444 | 0.497 | 0.356 | 0.347 |
| | Avg | **0.137** | 0.274 | 0.345 | 0.473 | 0.612 | 0.558 | 0.370 | 0.499 | 0.206 | 0.342 | 1.359 | 0.840 | 0.289 | 0.388 | 0.490 | 0.509 | 0.234 | **0.261** |

Table 3: Full results for the electricity load imputation task.

| Model | | PowerGPT | | PatchTST | | TS2Vec | | CoST | | TS-TCC | | TFC | | DLinear | | MICN | | LSTM | |
|---|---|---|---|---|---|---|---|---|---|---|---|---|---|---|---|---|---|---|---|
| Metric | | MSE | MAE | MSE | MAE | MSE | MAE | MSE | MAE | MSE | MAE | MSE | MAE | MSE | MAE | MSE | MAE | MSE | MAE |
| Exclusive | 0.125 | 0.364 | 0.395 | 0.382 | 0.446 | 0.327 | 0.354 | 0.623 | 0.568 | 0.265 | 0.314 | 0.330 | 0.372 | 0.332 | 0.357 | 0.263 | 0.338 | 0.904 | 0.629 |
| | 0.250 | 0.325 | 0.368 | 0.394 | 0.440 | 0.311 | 0.354 | 0.578 | 0.554 | 0.345 | 0.359 | 0.347 | 0.380 | 0.359 | 0.381 | 0.283 | 0.357 | 0.931 | 0.635 |
| | 0.375 | 0.330 | 0.376 | 0.392 | 0.440 | 0.312 | 0.348 | 0.557 | 0.548 | 0.323 | 0.354 | 0.370 | 0.384 | 0.385 | 0.402 | 0.306 | 0.379 | 1.034 | 0.667 |
| | 0.500 | 0.364 | 0.395 | 0.383 | 0.432 | 0.346 | 0.377 | 0.663 | 0.615 | 0.334 | 0.356 | 0.349 | 0.381 | 0.416 | 0.424 | 0.334 | 0.400 | 1.503 | 0.861 |
| | Avg | 0.345 | 0.384 | 0.388 | 0.439 | 0.324 | 0.359 | 0.605 | 0.571 | 0.317 | **0.346** | 0.349 | 0.379 | 0.373 | 0.391 | **0.296** | 0.368 | 1.093 | 0.698 |
| Public | 0.125 | 0.283 | 0.395 | 0.297 | 0.399 | 0.256 | 0.354 | 0.501 | 0.532 | 0.229 | 0.339 | 0.259 | 0.361 | 0.259 | 0.370 | 0.138 | 0.229 | 1.147 | 0.727 |
| | 0.250 | 0.258 | 0.368 | 0.293 | 0.394 | 0.244 | 0.354 | 0.443 | 0.507 | 0.260 | 0.357 | 0.256 | 0.363 | 0.285 | 0.391 | 0.150 | 0.244 | 1.189 | 0.742 |
| | 0.375 | 0.252 | 0.369 | 0.301 | 0.398 | 0.256 | 0.359 | 0.503 | 0.542 | 0.271 | 0.371 | 0.332 | 0.397 | 0.313 | 0.410 | 0.164 | 0.262 | 1.392 | 0.820 |
| | 0.500 | 0.283 | 0.395 | 0.302 | 0.399 | 0.264 | 0.369 | 0.564 | 0.587 | 0.276 | 0.371 | 0.271 | 0.375 | 0.344 | 0.432 | 0.178 | 0.278 | 1.987 | 0.984 |
| | Avg | 0.269 | 0.382 | 0.298 | 0.397 | 0.255 | 0.359 | 0.503 | 0.542 | 0.259 | 0.359 | 0.279 | 0.374 | 0.300 | 0.401 | **0.158** | **0.253** | 1.429 | 0.818 |
| Industry | 0.125 | 0.129 | 0.260 | 0.192 | 0.345 | 0.124 | 0.260 | 0.393 | 0.495 | 0.082 | 0.204 | 0.173 | 0.314 | 0.137 | 0.252 | 0.164 | 0.282 | 1.108 | 0.751 |
| | 0.250 | 0.122 | 0.255 | 0.142 | 0.295 | 0.112 | 0.250 | 0.400 | 0.507 | 0.117 | 0.248 | 0.172 | 0.312 | 0.161 | 0.283 | 0.178 | 0.302 | 1.174 | 0.768 |
| | 0.375 | 0.131 | 0.268 | 0.147 | 0.297 | 0.135 | 0.268 | 0.368 | 0.493 | 0.107 | 0.241 | 0.204 | 0.316 | 0.188 | 0.311 | 0.199 | 0.322 | 1.493 | 0.887 |
| | 0.500 | 0.129 | 0.260 | 0.159 | 0.308 | 0.136 | 0.278 | 0.457 | 0.557 | 0.115 | 0.241 | 0.151 | 0.299 | 0.216 | 0.338 | 0.219 | 0.345 | 2.155 | 1.098 |
| | Avg | 0.128 | 0.261 | 0.160 | 0.311 | 0.127 | 0.264 | 0.404 | 0.513 | **0.105** | **0.234** | 0.175 | 0.310 | 0.176 | 0.296 | 0.190 | 0.313 | 1.483 | 0.876 |
| District | 0.125 | 0.113 | 0.224 | 0.151 | 0.299 | 0.137 | 0.250 | 0.396 | 0.496 | 0.095 | 0.199 | 0.169 | 0.306 | 0.124 | 0.219 | 0.110 | 0.227 | 0.851 | 0.680 |
| | 0.250 | 0.116 | 0.228 | 0.138 | 0.278 | 0.123 | 0.257 | 0.377 | 0.481 | 0.128 | 0.241 | 0.200 | 0.307 | 0.152 | 0.258 | 0.113 | 0.239 | 0.834 | 0.678 |
| | 0.375 | 0.118 | 0.234 | 0.147 | 0.289 | 0.136 | 0.256 | 0.368 | 0.485 | 0.147 | 0.250 | 0.190 | 0.286 | 0.177 | 0.288 | 0.131 | 0.261 | 1.096 | 0.766 |
| | 0.500 | 0.113 | 0.224 | 0.154 | 0.292 | 0.140 | 0.276 | 0.529 | 0.601 | 0.128 | 0.238 | 0.153 | 0.277 | 0.206 | 0.317 | 0.143 | 0.274 | 1.858 | 1.062 |
| | Avg | **0.115** | **0.228** | 0.147 | 0.289 | 0.134 | 0.260 | 0.417 | 0.515 | 0.124 | 0.232 | 0.178 | 0.294 | 0.165 | 0.271 | 0.124 | 0.250 | 1.160 | 0.797 |
| City | 0.125 | 0.129 | 0.217 | 0.119 | 0.259 | 0.438 | 0.385 | 0.385 | 0.491 | 0.021 | 0.110 | 0.181 | 0.317 | 0.074 | 0.152 | 0.103 | 0.219 | 0.613 | 0.557 |
| | 0.250 | 0.104 | 0.192 | 0.128 | 0.315 | 0.094 | 0.205 | 0.137 | 0.301 | 0.031 | 0.136 | 0.173 | 0.295 | 0.093 | 0.191 | 0.116 | 0.230 | 0.777 | 0.617 |
| | 0.375 | 0.147 | 0.235 | 0.121 | 0.302 | 0.147 | 0.284 | 0.262 | 0.425 | 0.092 | 0.190 | 0.059 | 0.189 | 0.105 | 0.210 | 0.124 | 0.249 | 1.140 | 0.746 |
| | 0.500 | 0.129 | 0.217 | 0.102 | 0.247 | 0.135 | 0.266 | 1.089 | 0.877 | 0.042 | 0.139 | 0.443 | 0.428 | 0.123 | 0.234 | 0.201 | 0.318 | 1.789 | 1.073 |
| | Avg | 0.127 | 0.215 | 0.118 | 0.281 | 0.204 | 0.285 | 0.468 | 0.523 | **0.046** | **0.144** | 0.214 | 0.307 | 0.099 | 0.197 | 0.136 | 0.254 | 1.080 | 0.748 |
| Province | 0.125 | 0.076 | 0.178 | 0.079 | 0.248 | 0.939 | 0.488 | 0.409 | 0.516 | 0.033 | 0.142 | 0.373 | 0.438 | 0.064 | 0.134 | 0.061 | 0.177 | 0.342 | 0.454 |
| | 0.250 | 0.098 | 0.181 | 0.080 | 0.247 | 0.147 | 0.210 | 0.139 | 0.299 | 0.024 | 0.127 | 0.288 | 0.377 | 0.076 | 0.166 | 0.124 | 0.226 | 0.407 | 0.489 |
| | 0.375 | 0.070 | 0.174 | 0.088 | 0.255 | 0.299 | 0.382 | 0.216 | 0.388 | 0.094 | 0.180 | 0.038 | 0.158 | 0.085 | 0.185 | 0.084 | 0.210 | 0.577 | 0.564 |
| | 0.500 | 0.076 | 0.178 | 0.069 | 0.223 | 0.116 | 0.257 | 1.447 | 0.950 | 0.033 | 0.138 | 0.867 | 0.513 | 0.096 | 0.205 | 0.129 | 0.237 | 1.061 | 0.875 |
| | Avg | 0.080 | 0.178 | 0.079 | 0.243 | 0.375 | 0.334 | 0.553 | 0.538 | **0.046** | **0.147** | 0.392 | 0.371 | 0.080 | 0.172 | 0.100 | 0.213 | 0.597 | 0.596 |

Table 4: Full results for the electricity consumption imputation task.

| Model Metric | | PowerGPT MSE | PowerGPT MAE | PatchTST MSE | PatchTST MAE | TS2Vec MSE | TS2Vec MAE | CoST MSE | CoST MAE | TS-TCC MSE | TS-TCC MAE | TFC MSE | TFC MAE | DLinear MSE | DLinear MAE | MICN MSE | MICN MAE | LSTM MSE | LSTM MAE |
|---|---|---|---|---|---|---|---|---|---|---|---|---|---|---|---|---|---|---|---|
| Exclusive | 0.125 | 0.527 | 0.531 | 0.452 | 0.491 | 0.432 | 0.450 | 0.934 | 0.738 | 0.925 | 0.740 | 0.476 | 0.480 | 0.283 | 0.358 | 0.333 | 0.408 | 0.671 | 0.615 |
| | 0.250 | 0.495 | 0.516 | 0.467 | 0.502 | 0.433 | 0.455 | 0.827 | 0.694 | 0.945 | 0.749 | 0.472 | 0.476 | 0.303 | 0.377 | 0.353 | 0.427 | 0.674 | 0.616 |
| | 0.375 | 0.514 | 0.509 | 0.474 | 0.506 | 0.458 | 0.468 | 0.919 | 0.720 | 0.983 | 0.764 | 0.424 | 0.448 | 0.326 | 0.399 | 0.376 | 0.449 | 0.677 | 0.618 |
| | 0.500 | 0.425 | 0.466 | 0.502 | 0.520 | 0.479 | 0.478 | 0.885 | 0.714 | 1.036 | 0.782 | 0.472 | 0.474 | 0.355 | 0.421 | 0.405 | 0.471 | 0.693 | 0.627 |
| | Avg | 0.490 | 0.506 | 0.474 | 0.505 | 0.451 | 0.463 | 0.891 | 0.716 | 0.972 | 0.758 | 0.461 | 0.470 | **0.317** | **0.389** | 0.367 | 0.439 | 0.679 | 0.619 |
| Public | 0.125 | 0.442 | 0.506 | 0.290 | 0.376 | 0.322 | 0.388 | 0.853 | 0.692 | 0.897 | 0.739 | 0.396 | 0.434 | 0.158 | 0.248 | 0.208 | 0.298 | 0.539 | 0.576 |
| | 0.250 | 0.431 | 0.500 | 0.300 | 0.385 | 0.335 | 0.382 | 0.657 | 0.599 | 0.955 | 0.765 | 0.360 | 0.406 | 0.170 | 0.264 | 0.220 | 0.314 | 0.536 | 0.576 |
| | 0.375 | 0.424 | 0.473 | 0.309 | 0.388 | 0.361 | 0.407 | 0.895 | 0.706 | 0.996 | 0.765 | 0.259 | 0.339 | 0.184 | 0.281 | 0.234 | 0.332 | 0.572 | 0.572 |
| | 0.500 | 0.347 | 0.433 | 0.325 | 0.397 | 0.363 | 0.417 | 0.830 | 0.687 | 1.137 | 0.831 | 0.353 | 0.398 | 0.199 | 0.300 | 0.249 | 0.350 | 0.547 | 0.581 |
| | Avg | 0.411 | 0.478 | 0.306 | 0.386 | 0.345 | 0.399 | 0.809 | 0.671 | 0.996 | 0.775 | 0.342 | 0.394 | **0.178** | **0.273** | 0.228 | 0.323 | 0.539 | 0.576 |
| Industry | 0.125 | 0.284 | 0.401 | 0.305 | 0.425 | 0.290 | 0.401 | 0.921 | 0.758 | 0.904 | 0.754 | 0.352 | 0.444 | 0.184 | 0.302 | 0.234 | 0.352 | 0.579 | 0.631 |
| | 0.250 | 0.258 | 0.382 | 0.323 | 0.434 | 0.342 | 0.424 | 0.731 | 0.665 | 0.943 | 0.764 | 0.355 | 0.437 | 0.198 | 0.322 | 0.248 | 0.372 | 0.615 | 0.654 |
| | 0.375 | 0.279 | 0.399 | 0.330 | 0.440 | 0.356 | 0.442 | 0.856 | 0.716 | 1.028 | 0.797 | 0.312 | 0.399 | 0.219 | 0.342 | 0.269 | 0.392 | 0.619 | 0.654 |
| | 0.500 | 0.281 | 0.400 | 0.344 | 0.449 | 0.344 | 0.438 | 0.823 | 0.713 | 1.105 | 0.832 | 0.335 | 0.431 | 0.239 | 0.365 | 0.289 | 0.415 | 0.646 | 0.662 |
| | Avg | 0.275 | 0.396 | 0.325 | 0.437 | 0.333 | 0.426 | 0.833 | 0.713 | 0.995 | 0.787 | 0.339 | 0.428 | **0.210** | **0.333** | 0.260 | 0.383 | 0.615 | 0.650 |
| District | 0.125 | 0.124 | 0.269 | 0.244 | 0.365 | 0.255 | 0.364 | 1.024 | 0.826 | 0.898 | 0.752 | 0.348 | 0.441 | 0.130 | 0.247 | 0.180 | 0.297 | 0.601 | 0.615 |
| | 0.250 | 0.117 | 0.265 | 0.234 | 0.365 | 0.285 | 0.389 | 0.650 | 0.627 | 0.971 | 0.787 | 0.270 | 0.376 | 0.133 | 0.259 | 0.183 | 0.309 | 0.606 | 0.620 |
| | 0.375 | 0.156 | 0.305 | 0.272 | 0.387 | 0.320 | 0.419 | 0.927 | 0.752 | 0.987 | 0.779 | 0.270 | 0.352 | 0.151 | 0.281 | 0.201 | 0.331 | 0.635 | 0.635 |
| | 0.500 | 0.145 | 0.294 | 0.274 | 0.387 | 0.265 | 0.390 | 0.921 | 0.775 | 1.143 | 0.854 | 0.270 | 0.380 | 0.161 | 0.293 | 0.211 | 0.343 | 0.654 | 0.645 |
| | Avg | **0.136** | 0.283 | 0.256 | 0.376 | 0.281 | 0.391 | 0.880 | 0.745 | 1.000 | 0.793 | 0.285 | 0.387 | 0.143 | **0.270** | 0.194 | 0.320 | 0.624 | 0.629 |
| City | 0.125 | 0.100 | 0.246 | 0.223 | 0.348 | 0.266 | 0.372 | 0.939 | 0.802 | 1.051 | 0.822 | 0.270 | 0.370 | 0.123 | 0.239 | 0.173 | 0.289 | 0.422 | 0.522 |
| | 0.250 | 0.100 | 0.243 | 0.276 | 0.408 | 0.376 | 0.445 | 0.652 | 0.642 | 0.941 | 0.739 | 0.309 | 0.401 | 0.136 | 0.250 | 0.186 | 0.300 | 0.517 | 0.579 |
| | 0.375 | 0.142 | 0.297 | 0.217 | 0.346 | 0.307 | 0.411 | 0.869 | 0.712 | 1.009 | 0.766 | 0.360 | 0.411 | 0.144 | 0.269 | 0.194 | 0.319 | 0.516 | 0.579 |
| | 0.500 | 0.113 | 0.263 | 0.269 | 0.396 | 0.329 | 0.425 | 0.850 | 0.738 | 1.121 | 0.821 | 0.214 | 0.335 | 0.163 | 0.285 | 0.212 | 0.335 | 0.580 | 0.612 |
| | Avg | **0.114** | 0.263 | 0.246 | 0.375 | 0.319 | 0.413 | 0.827 | 0.723 | 1.030 | 0.787 | 0.288 | 0.379 | 0.141 | **0.261** | 0.191 | 0.311 | 0.509 | 0.573 |
| Province | 0.125 | 0.119 | 0.269 | 0.393 | 0.491 | 0.151 | 0.257 | 1.429 | 0.918 | 1.154 | 0.763 | 0.311 | 0.381 | 0.141 | 0.257 | 0.192 | 0.307 | 0.836 | 0.675 |
| | 0.250 | 0.113 | 0.256 | 0.307 | 0.419 | 0.492 | 0.485 | 0.893 | 0.722 | 0.893 | 0.686 | 0.439 | 0.457 | 0.204 | 0.306 | 0.254 | 0.356 | 0.759 | 0.640 |
| | 0.375 | 0.146 | 0.296 | 0.375 | 0.450 | 0.382 | 0.412 | 1.033 | 0.762 | 1.169 | 0.799 | 0.408 | 0.411 | 0.164 | 0.290 | 0.214 | 0.340 | 0.742 | 0.642 |
| | 0.500 | 0.135 | 0.283 | 0.335 | 0.433 | 0.427 | 0.414 | 1.016 | 0.759 | 1.125 | 0.794 | 0.298 | 0.389 | 0.194 | 0.309 | 0.244 | 0.359 | 0.707 | 0.621 |
| | Avg | **0.128** | **0.276** | 0.353 | 0.448 | 0.363 | 0.392 | 1.093 | 0.790 | 1.085 | 0.760 | 0.364 | 0.410 | 0.176 | 0.291 | 0.226 | 0.341 | 0.761 | 0.645 |

performs worse in this task (average F1-score). This may come from that anomaly detection requires the model to to find out the rare abnormal temporal semantic patterns and compare with other instance information, which the point-wise failed to represent the semantics in time series and lack of hierarchical information to refer.

Table 5: Full results for the electricity theft detection task. The P, R and F1 represent the precision, recall and F1-score (%) respectively. F1-score is the harmonic mean of precision and recall. A higher value of P, R and F1 indicates a better performance.

| Models | 1:10 | | | | 1:50 | | | | 1:100 | | | |
|---|---|---|---|---|---|---|---|---|---|---|---|---|
| | Pre. | Rec. | F1 | AUROC | Pre. | Rec. | F1 | AUROC | Pre. | Rec. | F1 | AUROC |
| DeepCNN | 0.092 | 0.516 | 0.156 | 0.502 | 0.020 | 0.592 | 0.039 | 0.510 | 0.011 | 0.302 | 0.020 | 0.509 |
| LSTM-FCN | 0.056 | 0.038 | 0.056 | 0.503 | 0.021 | 0.800 | 0.040 | 0.522 | 0.010 | 0.553 | 0.019 | 0.501 |
| TFC | 0.091 | 0.022 | 0.201 | 0.500 | 0.024 | 0.554 | 0.042 | 0.509 | 0.010 | 0.145 | 0.020 | 0.521 |
| PatchTST | 0.110 | 0.531 | 0.183 | 0.552 | 0.024 | 0.503 | 0.045 | 0.544 | 0.011 | 0.598 | 0.022 | 0.531 |
| TS2Vec | 0.102 | 0.613 | 0.175 | 0.538 | 0.025 | 0.293 | 0.045 | 0.530 | 0.014 | 0.148 | 0.025 | 0.520 |
| CoST | 0.115 | 0.326 | 0.170 | 0.538 | 0.020 | 0.312 | 0.037 | 0.504 | 0.012 | 0.139 | 0.022 | 0.512 |
| TS-TCC | 0.094 | 0.490 | 0.120 | 0.500 | 0.019 | 0.772 | 0.030 | 0.501 | 0.009 | 0.159 | 0.035 | 0.511 |
| PowerGPT | 0.136 | 0.604 | 0.212 | 0.578 | 0.038 | 0.890 | 0.057 | 0.599 | 0.033 | 0.501 | 0.066 | 0.557 |

**Miss Value Imputation.** We conduct miss value imputation on load and electricity consumption data with various randomly mask ratio $12.5\%, 25\%, 37.5\%, 50\%$. Due to the missing time points, the imputation task requires the model to discover underlying temporal patterns from the irregular and partially observed time series. We conduct load and electricity consumption imputation, the results are shown in Tab. 4 and Tab. 3 respectively. These results show that our proposed PowerGPT still achieves the consistent SOTA in these two task, demonstrating the advantages of leveraging both temporal and hierarchical view to enhance the prediction of the missing values.

## 5.1 MODEL ANALYSIS

**Low-resource labeled data evaluation.** In power systems, collecting series for downstream tasks is a significant investment. To demonstrate the practical application value of our work, we evaluate the performance of PowerGPT on downstream tasks under limited series availability. Specifically, the

Table 6: Full results for the elderly living alone detection task. The P, R and F1 represent the precision, recall and F1-score (%) respectively. F1-score is the harmonic mean of precision and recall. A higher value of P, R and F1 indicates a better performance.

| Models | 1:10 | | | | 1:50 | | | | 1:100 | | | |
|---|---|---|---|---|---|---|---|---|---|---|---|---|
| | Pre. | Rec. | F1 | AUROC | Pre. | Rec. | F1 | AUROC | Pre. | Rec. | F1 | AUROC |
| DeepCNN | 0.092 | 0.516 | 0.156 | 0.502 | 0.020 | 0.592 | 0.039 | 0.510 | 0.011 | 0.302 | 0.020 | 0.509 |
| LSTM-FCN | 0.056 | 0.038 | 0.056 | 0.503 | 0.021 | 0.800 | 0.040 | 0.522 | 0.010 | 0.553 | 0.019 | 0.501 |
| TFC | 0.091 | 0.022 | 0.201 | 0.500 | 0.024 | 0.554 | 0.042 | 0.509 | 0.010 | 0.145 | 0.020 | 0.521 |
| PatchTST | 0.110 | 0.531 | 0.183 | 0.552 | 0.024 | 0.503 | 0.045 | 0.544 | 0.011 | 0.598 | 0.022 | 0.531 |
| TS2Vec | 0.102 | 0.613 | 0.175 | 0.538 | 0.025 | 0.293 | 0.045 | 0.530 | 0.014 | 0.148 | 0.025 | 0.520 |
| CoST | 0.115 | 0.326 | 0.170 | 0.538 | 0.020 | 0.312 | 0.038 | 0.504 | 0.012 | 0.139 | 0.022 | 0.512 |
| TS-TCC | 0.094 | 0.490 | 0.120 | 0.500 | 0.019 | 0.772 | 0.030 | 0.501 | 0.009 | 0.159 | 0.035 | 0.511 |
| PowerGPT | 0.136 | 0.604 | 0.212 | 0.578 | 0.038 | 0.890 | 0.057 | 0.599 | 0.033 | 0.501 | 0.066 | 0.557 |

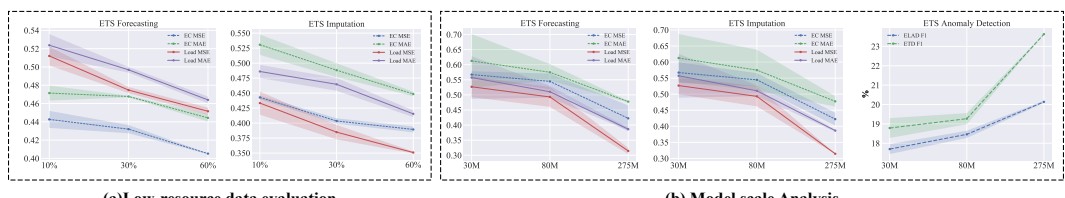

(a)Low-resource data evaluation      (b) Model scale Analysis

Figure 3: (a)Performance on all the downstream tasks across PowerGPTs with different ratio of finetuning dataset: 10%, 30% and 60%. (b)Performance on all the downstream tasks across PowerGPTs with different model size: PowerGPT-small (30M), PowerGPT-medium (80M) and PowerGPT (275M).

pre-trained model is finetuned on 10%, 30% and 60% of the set of finetuning dataset, respectively. After fine-tuning, models are evaluated on the same data which is sampled from the downstream dataset but non-overlapped with the finetuning data. As shown in Fig. 3(a), the performances of PowerGPT is decrease mildly when the finetuning data is fast decrease, demonstrating that PowerGPT fully captures the long-term temporal dependency and hierarchical correlation from the ETS data during pre-training and adapts to downstream taks more easily.

**Model scale Analysis.** To explore the impact of model size on performance, we additionally design three variants of PowerGPT with smaller size: PowerGPT-small, PowerGPT-medium, PowerGPT, and pre-train on the same datasets. All models are pre-trained with an Adam optimizer. PowerGPT-medium adopts the same learning rate as PowerGPT; for Power-small, the basic and maximum learning rate is $3e-6$. We evaluate these smaller size variants on all the downstream tasks. As shown in Fig. 3(b), as the model size increases, we observe an overall improvement in the performance of downstream tasks. Specifically, PowerGPT outperforms the other three variants across all metrics. Additionally, larger models exhibit decreased standard deviation, indicating more stable performance. The utilization of a larger model with higher capacity enables better generalization across a wide range of downstream tasks on such a huge ETS dataset.

# 6 CONCLUSION

In this paper, we propose a generic foundation model, PowerGPT, which learns powerful representations of electricity data. PowerGPT is the largest pre-training model on time series in power system, whose design (1) attends a long temporal dependency; (2) captures the hierarchical correlations across different instances. Experimentally, PowerGPT achieves consistent SOTA performance on various downstream tasks w.r.t. electrical scenarios. Further analysis shows the effectiveness and benefit of a large-scale pre-trained model in the field of electrical medicine. PowerGPT is an off-the-shelf model with its code and weights, which significantly alleviates the issue of sample and label efficiency and can directly participate in other electrical research.

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

## A    DATASET.

The original ETS data is provided by State Grid Corporation of China. After removing outliers, we sample a subset of this data, totaling 1TB in size, which includes electricity consumption and load data from 2016 to 2022, covering five levels: province, city, district, industry, and user. For pre-training dataset, we select the data from 2016 to 2020, while the data from 2020 onwards is utilized as the finetuning dataset for downstream tasks. Specifically, we apply sliding window $T_w$ on the ETS data of all instances to obtain a substantial number of records. In pre-training dataset, to alleviate the imbalance between electricity consumption and load data due to the different sample rate, we choose suitable size of stride for sliding. The detailed statistics of the pre-training dataset are summarized in Table 7. In finetuning dataset, in order to align with practical application scenarios, we don't adjust the stride to achieve data balance. In addition to ETS data, there is additional label information in the finetuning dataset, including labels for electricity theft users and elderly living alone. Among all users, about 4,000 users were confirmed as electricity thieve through on-site investigations by National Grid staff. Similarly, State Grid staff have also labeled 3,000 users as elderly living alone. These labels will serve as the ground-truth for two different downstream tasks, namely electricity theft detection and elderly living alone detection. The detailed statistics of the finetuning dataset is described in Table 8.

Table 7: Overall statistics of the pre-training dataset.

| | Instance | | Sampling Rate | Length | Stride | $T_w$ | # Records |
|---|---|---|---|---|---|---|---|
| Electricity Consumption | #province
#city
#region
#industry
#user | 1
11
90
45
1,530,826 | 1 day | 1826 | 1 | 256 | 1571
17,281
141,390
70,695
2,404,927,646 |
| Electricity Load | #province
#city
#region
#industry
#user | 1
11
90
45
1,530,826 | 15 minutes | 175,296 | 100 | 256 | 1751
19,261
157,590
78,795
2,680,476,326 |

Table 8: Overall statistics of datasets for downstream tasks.

| | Instance | | Anomaly Rate | Sampling Rate | Length | Stride | $T_w$ | # Records |
|---|---|---|---|---|---|---|---|---|
| Electricity Consumption Forecasting (Imputation) | #province
#city
#region
#industry
#user | 1
11
90
45
1,563,740 | – | 1 day | 730 | 1 | 256 | 475
5,225
42,750
21,375
742,776,500 |
| Electricity Load Forecasting (Imputation) | #province
#city
#region
#industry
#user | 1
11
90
45
1,563,740 | – | 15 minutes | 70,080 | 100 | 256 | 699
7689
220,680
110,340
3,753,585,352 |
| Electricity Theft Detection | #province
#city
#region
#industry
#user | 1
11
90
45
44,077 | –
–
–
–
9.1% | 1 day | 245,376 | 1 | 256 | 2452
26,972
220,680
110,340
108,076,804 |
| Elderly Living Alone Detection | #province
#city
#region
#industry
#user | 1
11
90
45
35,146 | –
–
–
–
8.5% | 1 day | 245,376 | 1 | 256 | 2452
26,972
220,680
110,340
86,177,992 |

**Pre-training.** For the model configurations, the temporal encoder contain a 16-layer Transformer encoder with model dimension 2048, inner dimension (FFN) 4096 and 16 attention heads, and the hierarchical encoder contains 2-layer R-GCN. During the pre-training, 40% patches in each

input sample are masked by learnable token in the form of temporal mask hierarchical mask. The model is pre-trained on a Linux system with 2 CPUs (AMD EPYC 9654 96-Core Processor) and 4 GPUs (NVIDIA Tesla A100 80G) for about 7.8 days. We optimize with Adam, updating the model parameters every 4 steps, and the model trains for 750k updates in total. A reduce learning rate on plateau scheduler is utilized to adjust learning rate during pre-training. Specifically, we set the basic learning rate as $1e-6$.

## B  BASELINES

Since we attempt to propose a foundation model in power systems for time series modeling, we extensively compare the well-acknowledged and advanced models in all four tasks, including supervised models: LSTM (Torres et al., 2022), LSTM-FCNs (Karim et al., 2019), CNN (Zheng et al., 2017)), DLinear (Zeng et al., 2023), MICN (Wang et al., 2022); self-supervised models: TS2Vec (Yue et al., 2022), TS-TCC (Eldele et al., 2021), TFC (Zhang et al., 2022b), CoST (Woo et al., 2022), PatchTST (Nie et al., 2023), which cover the majority of SOTA methods in task-specific models in power systems and general time series models.

## C  DOWNSTREAM TASKS

To verify the strong capacity of PowerGPT in modeling ETS data, we conduct extensive experiments on several downstream tasks, including forecasting and missing value imputation across different instances and different forecasting horizontal in electricity consumption and load data. Besides, electricity theft and elderly living alone detection are also involved. The detailed setups of these downstream tasks are as follows:

**Electricity Consumption and Load Data Forecasting.**  Predictive observation of the electricity data is beneficial for energy planning, market operations, trading decisions, and energy management. Therefore, we adopt short- and long-term ETS forecasting including electricity consumption and load data, in which the learned representations are fine-tuned to predict future series with different lengths given a past sequence. We set the past sequence to a fixed 256 points as the look-back window, and choose different lengths of predictive series for different tasks In electricity consumption data, the prediction lengths are set as {12, 48, 96} points. And in load data, the prediction lengths are set as {24, 96, 336, 720} points. A linear prediction head is used to predict the future signals. We adopt MAE and MSE as the performance metrics.

**Electricity Consumption and Load Data Imputation.**  During the process of collecting and recording power data, issues such as sensor malfunctions, communication failures, and manual entry errors often occur (Gomila & Clark, 2022), thus the power sequence recordings will be incomplete. Imputation can fill in these missing values so that subsequent data analysis and modeling can be performed based on complete sequences. For the imputation task, we randomly mask the timestamps in each patch with the ratio of {12.5%, 25.0%, 37.5%, 50.0%} and fine-tune the model to predict the missing values. We add a linear head to make predictions, then apply MAE and MSE as the evaluation metrics to measure the discrepancy between the masked and predicted values.

**Electricity theft detection.**  Electricity theft refers to the illegal operations by which users unauthorizedly tamper the electricity meter or wires to reduce or avoid consumption costs, and electricity thieves have a higher level of electrical power consumption with less stability (Hu et al., 2020). As one of the most important applications of electricity data modeling, electricity theft detection task is to evaluate the model ability to distinguish between the sequence of normal electricity consumption and the sequence of electricity theft incidents. We choose the electricity theft detection dataset provided by the Power Grid as the downstream dataset with about 10% positive (presence of electricity theft) samples. An MLP is adopted to classify the pre-trained representations. The evaluation metrics we use are accuracy, precision, recall, F1 scores.

**Elderly living alone detection.**  Increasingly serious aging of the population, the number of elderly people living alone is increasing. By conducting regular detect and identifying elderly living alone, communities and governments can enhance elderly care, improve the quality of life for those living alone, and contribute to smart aging initiatives. Zhang et al. (2022a) have find that the electricity consumption behavior of the elderly living alone is obvious periodic and highly

concentrated, so elderly living alone detection task is to differentiate between the normal household electricity consumption sequence and the elderly living alone electricity consumption sequence. We select elderly living alone dataset from Power Grid as the downstream dataset with about 10% positive (Elderly living alone) samples. An MLP is adopted to classify the pre-trained representations. And the evaluation metrics we use is like Electricity theft detection.

It is worth noting that in anomaly detection tasks, not all users have labels. Therefore, we treat those users as background nodes that are not used for loss computation but involved in the message passing of the GNN.

