# OpenReview forum: "PowerGPT: Foundation Model for Power Systems"
_ICLR.cc/2024/Conference — Submitted to ICLR 2024_

### Official Review · Reviewer_A7y6 · 2023-10-28

**Soundness:** 1 poor
**Presentation:** 2 fair
**Contribution:** 2 fair
**Rating:** 3
**Confidence:** 3

**Summary:**

Authors propose a foundation model to model electricity time series (ETS) data.
The objective is learning generic representations of electricity consumption data, providing a large-scale, off-the-shelf model for power systems.

**Strengths:**

The motivation and the idea behind the paper is interesting and the access to a big dataset of electricity time series is rare in this context and a foundation model off-the-shelf could be very interesting.

**Weaknesses:**

Generally speaking the work is not mature for a scientific publication and especially for ICLR2024
The literature review is insufficient, many recent and important trends were not taken in account.
The experimental results are not convincing and discussed wrongly.
An ablation study is not provided
The results are assessed considering only a portion of the considered dataset but no results are provided on literature dataset that can be useful to demonstrate how the trained model can be general.

**Questions:**

The authors talk about load and electricity consumption but they didn't describe the difference. Usually load/demand and consumption are used as synonyms (except some cases) and, if there is not the case, authors should describe clearly.
This is present over all paper and the authors should describe the difference and modify accordingly.

All the works and discussion related to the global models are missing.
For example : Montero-Manso, P.; Hyndman, R.J. "Principles and algorithms for forecasting groups of time series: Locality and globality", 2021; Buonanno et al., "Global vs. Local Models for Short-Term Electricity Demand Prediction in a Residential/Lodging Scenario", 2022; etc.

On Missing value imputation there is a lot of recent literature not considered.

"But most of them rely heavily on labeled data at scale, making it infeasible and expensive to obtain in power systems" --> authors should describe better.

Maybe there is a different understanding on labelling for timeseries but in forecasting or autoencoder-based models no labels are needed. What is the downstream task the authors have in mind when they talk about missing of labeled data? Also after the authors talk a lot about pre-training/fine-tuning and the necessity of the labelled data. For the forecasting, e.g., you can use transfer learning/fine-tune. This is related to the discussion on global models that is missing.

Related Works.
In Forecasting section: there are a lot of works, authors should at least cite as Makridakis competitions (M4, M5, M6), moreover, the Gradient Boosting methods (XGB and LGBM) are also often employed and not mentioned.

In missing value imputation section there are some recent works that use autoencoder methods as a fusion architecture that are not discussed [e.g., Pereira et al., Reviewing Autoencoders for Missing Data Imputation: Technical Trends, Applications and Outcomes, 2020; Buonanno et al.,Fusion of energy sensors with missing values, 2023; ]

Also in anomaly detection more recent works applied to the energy context are missing.

Fig.2(a) the colors of the nodes are too similar.

What is the meaning of "masked and unmasked patches of different nodes can overlap at temporal axis"? What is the unmasked patches? What is \tilde{N}?

What is the "learnable mask token"? A particular token that subtistute the missing data? How this token is learned?

The table 3 and 4 don't show that PowerGPT is SOTA for the imputation task!

A lot of typos:
e.g. empolyed
trys
we retrieval
horicontal
trianin
are in v --> are in bold?
Tab. ??
constrastive
to to
I suggest to carefully check the english

I suggest to reorganize the results section. In fact table 3 and 4 are discussed after table 5

Will the authors plan to release the trained models? The code? The dataset?

---

> ### Author Response · Authors · 2023-11-16
>
> Apologize for the poor writing and insufficient work, and add more __related works__.Thank you for your time in reviewing our paper and the detailed comments. We respond to some of the reviewer's comments:
> ***
> ***Q1: The difference between load and electricity consumption.***
>
> The sampling frequency of "load data" is 15 minutes, and its physical meaning is the total power consumption of various electrical devices borne by the user at a specific moment. The sampling frequency of "electricity consumption" is daily, and its physical meaning is the total electricity consumption by the user throughout the day.
>
> ***Q2: What is the meaning of "masked and unmasked patches of different nodes can overlap at temporal axis"? What is the unmasked patches? What is \tilde{N}?***
>
> Apologies for our poor expression. After careful review, we found that the correct wording for that section should be "at hierarchical axis" than "temporal axis".
>
> It means that for the hierarchical mask, it masks all the instance nodes' patches at the same moment, as shown by the gray mask in Figure 2(a). On the other hand, for the temporal mask, it randomly masks a certain number of patches along the temporal axis, as depicted by the blue mask in Figure 2(a). Therefore, different instance nodes have both masked and unmasked patches at the same moment, resulting in an overlap of these masked and unmasked patches.
>
> "\tilde{N}" represents the number of nodes sampled in the subgraph centered around the target node. In the text, it is discussed under the section __Randomly Masking__.
>
> ***Q3: What is the "learnable mask token"? A particular token that subtistute the missing data? How this token is learned?***
>
> The "learnable mask token" refers to using learnable parameters instead of setting the mask values to zero. It serves as a specific token that substitutes the missing data. This approach, as mentioned earlier, has been demonstrated and applied in [1,2]. By using learnable parameters for the mask, the model can better learn the representation of masked positions and improve the learning process.
>
> ***Q4: Will the authors plan to release the trained models? The code? The dataset?***
>
> Yes, we plan to release the code and the trained weights. And due to privacy, we couldn’t release the dataset.
>
> ## References ##
>
> [1] Cheng M, Liu Q, Liu Z, et al. TimeMAE: Self-Supervised Representations of Time Series with Decoupled Masked Autoencoders[J]. arXiv preprint arXiv:2303.00320, 2023.
>
> [2] He K, Chen X, Xie S, et al. Masked autoencoders are scalable vision learners[C]//Proceedings of the IEEE/CVF conference on computer vision and pattern recognition. 2022: 16000-16009.

---

> > ### Comment · Reviewer_A7y6 · 2023-11-22
> >
> > After reading other reviews, authors replies, and revised manuscript, I decided to maintain my initial score.

---

### Official Review · Reviewer_QMPP · 2023-10-31

**Soundness:** 2 fair
**Presentation:** 3 good
**Contribution:** 1 poor
**Rating:** 3
**Confidence:** 4

**Summary:**

This paper proposes a foundation model for electricity time series data called PowerGPT. The model is designed to explicitly model correlations across types of customers and hierarchies of aggregation levels. One such model is trained on a large dataset provided by State Grid Corporation of China spanning on ~1.5M user’s data spanning multiple years. The pretraining task is based on the masked autoencoder strategy. It is empirically compared to a variety of deep SOTA time series models on pretrain-then-fine tasks using the same State Grid data (forecasting, anomaly detection, and missing value imputation).

**Strengths:**

- I believe studying the application of the ideas behind foundation models to energy systems to be of key importance.
- The PowerGPT architecture combines key ideas (temporal patching, random masking, and hierarchical GNNs) in a sound way.
- The empirical results on multiple downstream tasks validate the effectiveness of the architecture on the State Grid dataset.
- I think it is valuable to provide evidence that relatively large transformers are able to be trained on large, diverse time series datasets (see [1,2]).

**Weaknesses:**

- Overall I believe the significance and novelty of this paper is low. Despite the large size of the model, which is interesting, this by itself is not a sufficiently significant or novel contribution for ICLR.
- Moreover, I do not think the problem setting is of wide interest, as I am confident that geographical electricity time series data, down to the level of specific users linked on a graph, in actual cities is not in general publicly available data (see possible ethical concerns).
- The model is only evaluated on the State Grid test datasets/tasks, which is a dataset introduced by the authors and which has not been vetted by peer review. It would be recommended to conduct experiments on one or more established benchmarks as well. I can recommend BuildingsBench [2], which is a recently published benchmark of 7 electricity datasets for evaluating pretraining-the-finetuning approaches for short-term (24 hour) building load forecasting.
- A discussion on related work is missing. See examples of references, including papers on transformers for load forecasting as well as transfer learning for load forecasting [3,4,5,6].
- The paper needs proofreading to correct typos and fix grammar issues.


### References

1. Kunz, Manuel, Stefan Birr, Mones Raslan, Lei Ma, Zhen Li, Adele Gouttes, Mateusz Koren et al. "Deep Learning based Forecasting: a case study from the online fashion industry." arXiv preprint arXiv:2305.14406 (2023).
2. Emami, Patrick, Abhijeet Sahu, and Peter Graf. "BuildingsBench: A Large-Scale Dataset of 900K Buildings and Benchmark for Short-Term Load Forecasting." NeurIPS 2023 Datasets & Benchmarks. https://arxiv.org/abs/2307.00142
3. Chadoulos, Spiros, Iordanis Koutsopoulos, and George C. Polyzos. "One model fits all: Individualized household energy demand forecasting with a single deep learning model." In Proceedings of the Twelfth ACM International Conference on Future Energy Systems, pp. 466-474. 2021.
4. Hertel, Matthias, Simon Ott, Oliver Neumann, Benjamin Schäfer, Ralf Mikut, and Veit Hagenmeyer. "Transformer Neural Networks for Building Load Forecasting." In Tackling Climate Change with Machine Learning: workshop at NeurIPS 2022. 2022.
5. He, Yu, Fengji Luo, and Gianluca Ranzi. "Transferrable model-agnostic meta-learning for short-term household load forecasting with limited training data." IEEE Transactions on Power Systems 37, no. 4 (2022): 3177-3180.
6. Xu, Xianze, and Zhaorui Meng. "A hybrid transfer learning model for short-term electric load forecasting." Electrical engineering 102 (2020): 1371-1381.

**Questions:**

- Why did the authors make a distinction between electricity consumption (1 day temporal granularity) and electricity load (15 minute)?
 It seems like 1 day forecasts could be obtained by aggregating the predictions made at a finer granularity?

**Details Of Ethics Concerns:**

I have noticed the following which I believe may constitute an ethical concern.

- The State Grid dataset introduced in this paper includes “fine-grained user-related information”. It is not made clear what exactly this entails and whether this includes personally identifiable information (PII)? Also did, the "users" consent to have this information shared and used for this purpose?
- The tasks “electricity theft detection” and “elder living alone” detection appear to be intended to evaluate the capabilities of this foundation model to surveil individuals. Clearly, this PowerGPT model is "dual-use" in the sense that it can be misused for surveillance. This is potentially concerning.
- As an aside, those outside of China may not feel comfortable clicking a “.com.cn” domain URL to view the data.

---

> ### Author Response · Authors · 2023-11-16
>
> Thanks for your time and effort in reviewing our paper. Sorry for the typos and grammar issues, we will check and fix these issues carefully.The following are our responses to your specific questions and concerns:
> ***
> ***Q1: About ethical concerns.***
>
> All our data and tasks have received ethical approval, and all data related to personal information have undergone de-identification processing. The fine-grained information only consists of electricity consumption curves at the user level, with no personally identifiable information (PII) involved.
>
> From our understanding, deep learning models can be used to detect user anomalous behavior in various domains, including financial fraud[1,2] and telephone scams[3]. Platforms like Twitter and Instagram also utilize different user behavior patterns to customize recommendations for their users[4,5].These models do not surveil users; they simply classify based on user behavior.
>
> Just like the aforementioned tasks, the tasks “electricity theft detection” and “elder living alone” detection also involve the use of electricity consumption sequences for "anomaly detection" purposes. These tasks have undergone national ethical review to ensure their compliance with ethical standards.
>
>
> ***Q2: It would be recommended to conduct experiments on one or more established benchmarks as well.***
>
> Thanks for your advices, in the future we will onduct extensive experiments on other publicly available datasets.
>
> ***Q3: Why did the authors make a distinction between electricity consumption (1 day temporal granularity) and electricity load (15 minute)?***
>
> The sampling frequency of the load data is 15 minutes, and its physical significance is the total power consumption of various electrical devices that a user bears at a specific moment. On the other hand, the sampling frequency of electricity consumption data is per day, and its physical significance is the total electricity usage by a user throughout a day.
>
> - Based on the actual requirements, the maximum prediction window for load data is 7 days, approximately 700 points, while for electricity consumption, the prediction window can extend up to three months with around 90 points. The prediction windows for load and electricity consumption are inconsistent.
> - Not all users have both load and electricity consumption data available in the raw data.
>
> So we make a distinction between electricity consumption (1 day temporal granularity) and electricity load (15 minute), and use them separately for downstream tasks.
>
>
> ## References ##
>
> [1] Ngai E W T, Hu Y, Wong Y H, et al. The application of data mining techniques in financial fraud detection: A classification framework and an academic review of literature[J]. Decision support systems, 2011, 50(3): 559-569.
>
> [2] Liu Y, Ao X, Qin Z, et al. Pick and choose: a GNN-based imbalanced learning approach for fraud detection[C]//Proceedings of the web conference 2021. 2021: 3168-3177.
>
> [3] Xing J, Yu M, Wang S, et al. Automated fraudulent phone call recognition through deep learning[J]. Wireless Communications and Mobile Computing, 2020, 2020: 1-9.
>
> [4] Godin F, Slavkovikj V, De Neve W, et al. Using topic models for twitter hashtag recommendation[C]//Proceedings of the 22nd international conference on world wide web. 2013: 593-596.
>
> [5] Guo D, Xu J, Zhang J, et al. User relationship strength modeling for friend recommendation on Instagram[J]. Neurocomputing, 2017, 239: 9-18.

---

> > ### Comment · Reviewer_QMPP · 2023-11-22
> >
> > I have read the author response, other reviews, and the revised manuscript. I have decided to maintain my initial score.

---

### Official Review · Reviewer_fens · 2023-10-31

**Soundness:** 3 good
**Presentation:** 3 good
**Contribution:** 2 fair
**Rating:** 5
**Confidence:** 4

**Summary:**

The authors propose a time series model for predicting electricity consumption using a large amount of data from a province in China. Their model is a transformer-based model that takes into account not only the temporal dimension of the problem but also the hierarchical structure of the power network. Their model works very well on several downstream tasks involving forecasting, missing value imputation, and anomaly detection.

**Strengths:**

- The proposed PowerGPT model works very well in forecasting power consumption across different horizons, beating other state-of-art time-series forecasting models.

- In addition to the usual time series window, the model take into account the hierarchical relations in a power network to help with forecasting.

- The model also excels in other downstream tasks such as missing value imputation and anomaly detection.

**Weaknesses:**

- We do see significant improvements in forecasting performance by PowerGPT. However, it does use more information, especially the hierarchical information in the forecasting. It is not clear from the paper how those information are used. Do we get the same history window from different hierarchies? For example, when predicting at the district level, do we get the past 256 history values at city, province, industry and user level?

- Although the authors claim that their PowerGPT is a foundational model, there are no transfer experiments on other datasets. We don't know how well the model might perform on electricity consumption data from other countries, or even a different province in China (since the dataset only contain 1 province). This makes the claim of a foundational model weak.

- From the results in the tables I cannot see any forecasting experiments on individual user electricity load. These time series are usually much more variable and interesting. are they indicated by one of the rows 'exclusive' or 'public' in the tables? There are no explanations for those.

**Questions:**

- Why is a history window of size 256 used? For 15-min user level data that's not even 1 week of consumption data to capture the weekly patterns. Have the authors tried longer windows sizes like 512 or 1024?

Minor typos:
- p6, Table labels are missing
- p8, 'miss value' -> 'missing value'

**Details Of Ethics Concerns:**

The electricity consumption data at individual household level are private data. Need to make sure that the data is not shared together with the code.

---

> ### Author Response · Authors · 2023-11-16
>
> We thank the reviewer for their time and feedback on our paper. We respond to some of the reviewer's comments:
>
> And we apologize for the poor writing. We will rewrite most of the content and carefully check the spelling. We are sorry for any inconvenience caused to your reading and review.
> ***
> ***Q1: For example, when predicting at the district level, do we get the past 256 history values at city, province, industry and user level?***
>
> Yes,when predicting an instance of some level, we will get other level instances’ history values.
>
> When predicting a specific instance, such as a district, we simultaneously obtain a subgraph centered around that node. The subgraph includes its high-level instance: city and province, as well as its low-level instance: users. Each node contains a historical sequence of 256 points for the same time period. As shown in Figure 2 (during downstream tasks, there is no masking process), R-GCN models the connectivity between different instances in the subgraph, with different edges having distinct labels and weights. Therefore, when predicting the future sequence based on the past 256 points of a particular node, PowerGPT takes into account the values of the past 256 points of the node's high and low level nodes.
>
>
> ***Q2:there are no transfer experiments on other datasets.***
>
> Sorry, due to certain limitations, we have not yet obtained electricity datasets for other provinces or cities. In our future work, we plan to incorporate electricity data from additional regions to conduct more comprehensive transfer experiments.
>
> ***Q3: Are individual user indicated by one of the rows 'exclusive' or 'public' in the tables?***
>
> In the power system, residential users are divided into two categories: Exclusive Transformer (ET) users and Public Transformer (PT) users. ET users are primarily large consumers from industrial, agricultural, office, and commercial sectors. They exhibit distinct electricity consumption patterns, with higher electricity usage and load values. ET users have their own transformers, which are invested in and established by the users themselves. On the other hand, most PT users are residential customers, and their transformers are invested in and established by the power companies to provide public electricity supply.
>
> Classifying and predicting residential users in the power system has significant economic benefits. Therefore, in the paper, we present the results by categorizing users into 'exclusive' or 'public' groups separately.
>
> ***Q4: Why is a history window of size 256 used?***
>
> According to the data we received, the number of points in the load data for the same number of days is 96 times higher than the electricity consumption data (due to the sampling frequency of the load being 96 times higher). As a result, when increasing the training window length, the amount of training data for electricity consumption decreases. To maintain a relatively balanced proportion of training data, we set the window to 256.
>
> However, this may not be an optimal choice. In the future, we plan to increase the window length while performing upsampling to ensure an adequate amount of training data for electricity consumption. Additionally, we will also explore the use of non-uniform training lengths to enhance the robustness of the model.

---

### Official Review · Reviewer_mCC2 · 2023-11-01

**Soundness:** 2 fair
**Presentation:** 2 fair
**Contribution:** 1 poor
**Rating:** 1
**Confidence:** 4

**Summary:**

The paper proposes PowerGPT, a foundation model for electricity time series (ETS) data in power systems. PowerGPT is pre-trained on a large-scale ETS dataset and captures long-term temporal dependency and hierarchical correlation. It achieves state-of-the-art performance on various downstream tasks in power systems, such as forecasting, missing value imputation, and  anomaly detection. The paper highlights the effectiveness of the large-scale pre-training strategy and explores the impact of model size on performance.

**Strengths:**

1. Foundation model for power systems could be a powerful assistant tool for dispatcher. However, whether it should be used for forecasting and anomaly detection is worthy for more discussion.

**Weaknesses:**

1. ETS data is not as structural as language tokens, which actually reflects the inherent physical laws of power systems, as well as human behaviors. These can hardly be captured by foundation model and also be predicted by autoregressive methods.
2. The completion level of the article is low, and the dataset is not open-source. There are multiple referencing errors.

**Questions:**

1. Can PowerGPT adapt to topology changes without re-training?
2. Can you demonstrate performance on open-sourced dataset? Such as Pecan Street.

---

> ### Author Response · Authors · 2023-11-16
>
> We thank the reviewer for the constructive feedback and provide responses below.
>
> And we apologize for the poor writing. We will rewrite most of the content and carefully check the spelling. We are sorry for any inconvenience caused to your reading and review.
> ***
> ***Q1: The feasibility of modeling and predicting ETS.***
>
> Although ETS data is not as structured as language tokens, it shares inherent patterns of variation with any other time series in different domains. Nowadays, there are various works[1-9] dedicated to modeling and predicting time series data in publicly available datasets such as ETT, WEATHER, ILLNESS, ELECTRICITY, and CHARGE, covering domains like finance, disease, energy and weather. __And these models all use transformer to model time series__. Especially after patching, the time token is simialr with the language token in NLP, which is the embedding including the neighbor imformation.
>
> These works demonstrate the feasibility of time series modeling and highlight their effectiveness in various tasks such as time series forecasting, classification, and anomaly detection. They provide evidence of the potential and usefulness of time series analysis in these domains.
>
> ***Q2: Can PowerGPT adapt to topology changes without re-training?***
>
> PowerGPT can adapt to the changes in the topology of downstream tasks. In various downstream tasks, such as those influenced by population mobility or changes in regional divisions, the instances forming the topology can vary significantly. Graph Neural Networks (GNNs) excel at modeling such changing topologies, which does not affect the prediction of single node due to changes in the overall topology.
>
> ***Q3: Can you demonstrate performance on open-sourced dataset? Such as Pecan Street.***
>
> Sorry, We are in the process of applying for access to the Pecan Street dataset, and once our application is approved, we will proceed with testing on the open-source dataset.
>
> ## References ##
>
> [1] Zhou H, Zhang S, Peng J, et al. Informer: Beyond efficient transformer for long sequence time-series forecasting[C]//Proceedings of the AAAI conference on artificial intelligence. 2021, 35(12): 11106-11115.
>
> [2] Wu H, Xu J, Wang J, et al. Autoformer: Decomposition transformers with auto-correlation for long-term series forecasting[J]. Advances in Neural Information Processing Systems, 2021, 34: 22419-22430.
>
> [3] Zhou T, Ma Z, Wen Q, et al. Fedformer: Frequency enhanced decomposed transformer for long-term series forecasting[C]//International Conference on Machine Learning. PMLR, 2022: 27268-27286.
>
> [4] Yue Z, Wang Y, Duan J, et al. Ts2vec: Towards universal representation of time series[C]//Proceedings of the AAAI Conference on Artificial Intelligence. 2022, 36(8): 8980-8987.
>
> [5] Wu H, Hu T, Liu Y, et al. TimesNet: Temporal 2D-Variation Modeling for General Time Series Analysis[C]//The Eleventh International Conference on Learning Representations. 2022.
>
> [6] Nie Y, Nguyen N H, Sinthong P, et al. A Time Series is Worth 64 Words: Long-term Forecasting with Transformers[C]//The Eleventh International Conference on Learning Representations. 2022.
>
> [7] Wang H, Peng J, Huang F, et al. Micn: Multi-scale local and global context modeling for long-term series forecasting[C]//The Eleventh International Conference on Learning Representations. 2022.
>
> [8] Zhang Y, Yan J. Crossformer: Transformer utilizing cross-dimension dependency for multivariate time series forecasting[C]//The Eleventh International Conference on Learning Representations. 2022.
>
> [9] Cheng M, Liu Q, Liu Z, et al. TimeMAE: Self-Supervised Representations of Time Series with Decoupled Masked Autoencoders[J]. arXiv preprint arXiv:2303.00320, 2023.

---

### Meta-Review · Area_Chair_Ky7M · 2023-12-10

**Metareview:**

While the reviewers appreciated the problem the paper studies and the soundness of the writing and experiments, they had concerns with (a) the overall contribution of the work, (b) the lack of dicussion on related work, (c) claims made about the model being a foundation model without experiments showing that the model transfers to other datasets, (d) inexact language which caused a number of confusions. The sheer number of outstanding reviewer concerns make it unclear if the paper is able to be revised to address them all for this reviewing cycle. Given these open points, I believe this work should be rejected at this time. Once these things are addressed in an updated version the work will be much improved.

**Justification For Why Not Higher Score:**

There were just too many concerns to be fixed within a rebuttal period, the authors could not respond to all of them adequately.

**Justification For Why Not Lower Score:**

N/A

---

### Decision · Program_Chairs · 2024-01-16

Reject